# The Influence of Medium on Fluorescence Quenching of Colloidal Solutions of the Nd^3+^: LaF_3_ Nanoparticles Prepared with HTMW Treatment

**DOI:** 10.3390/nano12213749

**Published:** 2022-10-25

**Authors:** Elena Timofeeva, Elena Orlovskaya, Alexandr Popov, Artem Shaidulin, Sergei Kuznetsov, Alexandr Alexandrov, Oleg Uvarov, Yuri Vainer, Gleb Silaev, Mihkel Rähn, Aile Tamm, Stanislav Fedorenko, Yurii Orlovskii

**Affiliations:** 1Prokhorov General Physics Institute of the Russian Academy of Sciences, Vavilov Str. 38, 119991 Moscow, Russia; 2Institute of Spectroscopy of the Russian Academy of Sciences, Fizicheskaya Str. 5, Troitsk, 108840 Moscow, Russia; 3Higher School of Economics, National Research University, Myasnitskaya Str. 4, 101000 Moscow, Russia; 4Institute of Physics, University of Tartu, W. Ostwaldi Str. 1, 50411 Tartu, Estonia; 5Voevodsky Institute of Chemical Kinetics and Combustion SB RAS, Institutskaya Str. 3, 630090 Novosibirsk, Russia

**Keywords:** aqueous colloidal solutions of Nd^3+^: LaF_3_ nanocrystals, hydrothermal microwave (HTMW) treatment, DMSO solvent, NIR fluorescence kinetics, static fluorescence quenching on OH- acceptors, relative fluorescence quantum yield, colloidal clusters, ultramicroscope, nanoparticle tracking analysis (NTA)

## Abstract

An original method was proposed to reduce the quenching of the NIR fluorescence of colloidal solutions of 0.1 at. % Nd^3+^: LaF_3_ nanoparticles (NPs) synthesized by aqueous co-precipitation method followed by hydrothermal microwave treatment. For this, an aqueous colloidal solution of NPs was precipitated by centrifugation and dissolved in the same volume of DMSO. The kinetics of static fluorescence quenching of Nd^3+^ donors of doped NPs dispersed in two solvents was analyzed to determine and to compare the concentrations of OH- quenching acceptors uniformly distributed throughout the volume of the NPs. The dependences of the relative fluorescence quantum yield *φ* of colloidal solutions on the concentration of OH- groups in the NPs were calculated and were also used to determine concentration of acceptors in the volume of NPs in different solvents. It was found that the concentration of OH- groups in NPs dispersed in DMSO is almost two times lower than in NPs dispersed in water. This gives an almost two-fold increase in the relative fluorescence quantum yield *φ* for the former. The sizes of synthesized NPs were monitored by common TEM and by applying a rapid procedure based on optical visualization of the trajectories of the Brownian motion of NPs in solution using a laser ultramicroscope. The use of two different methods made it possible to obtain more detailed information about the studied NPs.

## 1. Introduction

Fundamental research on luminescence of colloidal nanoparticles (NPs) is important due to the prospects for their wide application in different fields of biological and medical applications, such as bioimaging, local controlled heating, local thermal destruction, and drug delivery. Doping the NPs with rare earth ions (REI), for example, with the Nd^3+^ ions, opens new perspectives for their application in bioimaging. These ions have absorption and luminescence spectral lines in the first biological window in the spectral range of 750–950 nm, i.e., in the near infrared (NIR). At present, an urgent problem is the creation of efficient nontoxic colloidal solutions of nanosized phosphors based on dielectric crystals doped with REIs. Such luminescent NPs have a number of unique properties, such as narrow absorption and luminescence spectral lines, long lifetimes of excited states—which allow time detuning from biotissue autofluorescence, and high photo- and physico-chemical stability [1,2,3,4,5]. They can be used in biology and medicine as in vitro and in vivo luminescent probes for visualization in the visible (VIS) and near infrared (NIR) spectral ranges in the first (0.75–0.95 µm) [6,7], second (1.0–1.2 µm) [8] and third (1.3 µm) biological windows [9]. Bioimaging using NIR radiation is preferable over visible light, since it penetrates the biological tissue to a depth of one or more centimeters [10,11,12,13,14,15]), without causing photoinduced cytotoxicity.

One of the problems that arise when using nanoparticles synthesized by aqueous methods is the presence of structural OH- groups and water in mesopores, which leads to quenching of the luminescence of RE doping ions. The simplest way to remove water is to dry the NPs at an appropriate temperature; however, this method cannot be called universal and suitable for use for any purpose, since it is important for medical research that the NPs be in a stable nontoxic colloidal solution. In the case of aqueous colloidal solutions of the NPs, the maximum possible removal of water from NPs can be achieved using a hygroscopic liquid that will not affect the NPs in any way (i.e., will not react with them), will be miscible with water in any proportions, and will not be toxic. Based on the foregoing, dimethyl sulfoxide (CH_3_)_2_SO (DMSO) was chosen as a “drying agent”, a polar aprotic universal solvent for organic and inorganic compounds, resistant to oxidation and reduction processes. DMSO is widely used in chemistry as a universal aprotic solvent, and in medicine as an antiseptic and anti-inflammatory agent, and meets all of the listed mandatory criteria [16,17,18,19]. It will not be superfluous to note a number of properties of dimethyl sulfoxide, some of which are quite unique.

DMSO has the ability to penetrate biological membranes (such as skin and mucous membranes) without damaging them, increasing the transdermal transfer of substances dissolved in it, penetrating even through the endothelial coatings of the walls of blood vessels and the brain, i.e., they can even overcome the blood-brain (encephalic) barrier (a semi-permeable barrier between blood and nervous tissue that prevents large or polar molecules, as well as blood cells, from entering the brain), which is inaccessible to conventional drug therapy.DMSO exhibits transparency in the spectral range of 350–2200 nm (according to some sources from 260 nm).DMSO is a good blood and tissue preservation agent.

Regarding the toxicity of DMSO, there are still conflicting opinions in the literature, but most of the claims made have no real evidence. For example, in 1965, studies of DMSO were banned due to the alleged appearance of changes in the lens of the eye of experimental animals, which was later refuted by preclinical and clinical studies in animals and humans [16,17,18,19,20].

In this work, we study and compare the fluorescence quenching of aqueous colloidal solutions of Nd^3+^: LaF_3_ NPs with the same NPs redispersed in DMSO.

Another problem of using of NPs synthesized by colloidal synthesis in medical and other applications is their aggregation into clusters [21,22,23]. This process can occur both during the synthesis of NPs and during storage of their solutions. The control of these processes using an electron microscope is a complicated and lengthy procedure and requires drying of the sample that may distort the information obtained as a result of possible changes in the structure of the clusters and their additional aggregation during drying.

This paper reports the results of applying a simple and rapid control method based on the nanoparticle tracking analysis (NTA) of Brownian motion of single NPs in solutions. To visualize these trajectories, we have developed a highly sensitive laser ultramicroscope operating in the mode of recording scattering signals at an unshifted frequency. The essential advantage of this method is the visualization and analysis of individual trajectories of single particles that allows inherent problems associated with averaging over an ensemble of particles to be overcome. For example, such averaging is inherent for another method widely used for diagnostics of NPs in solutions: the dynamic light scattering (DLS). The latter is based on measurement of the total light scattering of ensemble NPs in solution and yields information averaged over an ensemble of particles, which is usually heavily weighted towards small values of large, usually contaminant, particles. The use of two different methods here made it possible to obtain more detailed information about the studied NPs.

## 2. Quantum Yield of Static Fluorescence Quenching

A convenient quantitative criterion for the degree of luminescence quenching is the relative luminescence quantum yield, which is determined from the measured luminescence kinetics of the excited electronic level as presented in [24].
(1)$$\varphi =\frac{1}{\tau_{R}}\int_{0}^{\infty }I_{meas}(t)dt=\frac{1}{\tau_{R}}\int_{0}^{\infty }N(t)\exp\left(-t/\tau_{R}\right)dt$$

Here, $\tau_{R}$ is the radiative lifetime of the donor, and N(t) is the kinetics of impurity luminescence quenching. If the concentration of donors is low enough, the electronic excitation is quenched at the initially excited donor. This process is called static quenching, since the excitation does not migrate over the donors, but “dies” at the site of its origin. As a consequence, the quenching kinetics $N(t)=N_{0}(t)$ depends on the concentration of acceptors $n_{A}$ and the corresponding probability of a nonradiative transfer of an electronic excitation to an acceptor w(r) as follows [24,25].
(2)$$N_{0}(t)=\exp\{-n_{A}4\pi \int_{0}^{\infty }drr^{2}[1-e^{-w(r)t}]\}$$

For the dipole–dipole mechanism of donor–acceptor interaction,
(3)$$w(r)=\frac{C_{DA}}{r^{6}}=\frac{1}{\tau_{R}}{\left(\frac{R_{F}}{r}\right)}^{6}$$
the kinetics of static quenching $N_{0}(t)$ of an excited donor by a group of acceptors randomly distributed in the bulk has a well-known square root Förster kinetics [26].
(4)$$N_{0}(t)=\exp\left(-\gamma_{A}\sqrt{t}\right)$$

Here the macroparameter of static quenching is determined as follows
(5)$$\gamma_{A}=\frac{4}{3}\pi^{3/2}n_{A}\sqrt{C_{DA}}=\frac{4}{3}\pi R_{F}^{3}n_{A}\sqrt{\frac{\pi }{\tau_{R}}}$$

Substituting (3) into (1) and integrating, we obtain the concentration dependence of the relative luminescence quantum yield:(6)$$\varphi {|}_{n_{D}\to 0}=1-f\left(\gamma_{A}\sqrt{\tau_{R}}/2\right);\begin{array}{cc} & f(x)=\sqrt{\pi }xe^{x^{2}} \end{array}erfc(x)$$

Here erfc(x) is the complementary error function [27]. As follows from (6), the quantum yield in the static case is a non-linear decreasing function of one universal variable x, which is a composite quantity depending on the product $\gamma_{A}\sqrt{\tau_{R}}$. By definition, this value is proportional to the number of acceptors in the volume of a sphere of Förster radius $R_{F}$
(7)$$x=\gamma_{A}\sqrt{\tau_{R}}/2=\frac{\sqrt{\pi }}{2}\frac{4\pi }{3}R_{F}^{3}n_{A}$$

When the argument x changes from zero to infinity, the quantum yield (6) changes from one to zero (Figure 1). For small values of the argument, when the quenching is weak, the quantum yield is close to unity and decreases linearly
(8)$$\varphi =1-\sqrt{\pi }x=1-\sqrt{\pi \tau_{R}}\gamma_{A}/2;\begin{array}{cc} & x<<1 \end{array}$$
while for large values of the argument (strong quenching, low quantum yield), the change occurs inversely proportionally to the square of the argument
(9)$$\varphi =\frac{1}{2x^{2}}=\frac{2}{\gamma_{A}^{2}\tau_{R}};\begin{array}{cc} & x>>1 \end{array}$$

Note that the asymptotic Formula (9) is obtained from (1) and (4) neglecting radiative decay against the background of rapidly decaying static kinetics ($\exp\left(-t/\tau_{R}\right)\approx 1$).

## 3. Materials and Methods

### 3.1. Synthesis of the Studied Aqueous Colloids of the LaF_3_: Nd^3+^ Nanoparticles

A colloidal solution of 0.1 at. % Nd^3+^: LaF_3_ NPs was synthesized from an aqueous solution by the hydrothermal method with microwave treatment (HTMW). Its detailed characterization including X-ray phase analysis of the synthesized LaF_3_ NPs was carried out in Ref. [28] (see Supplementary Materials). Next, this colloidal aqueous solution of NPs was divided into two equal parts, one part of the NPs was precipitated by centrifugation and dispersed in the same volume of DMSO (the condition was strictly observed that the volume concentration of NPs in both colloid samples was the same—20 mg/mL). As a result, two aqueous colloidal solution samples were obtained: one of 0.1 at. % Nd^3+^: LaF_3_ NPs (ACS), and one of 0.1 at. % Nd^3+^: LaF_3_ NPs in a DMSO dispersion medium (DMSO). The DMSO colloid was then centrifuged again and redispersed in water to obtain a reduced aqueous colloidal solution of LaF_3_ NPs (DMSO-ACS). Finally, the last sample was again subjected to the procedures described above and the nanoparticles were again transferred into a new DMSO solution, followed by another redispersion step in a fresh DMSO solution (we designate this sample as ACS-DMSOx2).

### 3.2. Morphology of Synthesized Nanoparticles

Visualization of the morphology of synthesized 0.1 at. % Nd^3+^: LaF_3_ NPs and their size distribution characterization were carried out at the Institute of Physics of the University of Tartu (Estonia) using transmission electron microscopy (TEM) analysis. The measurements were performed in the scanning mode (STEM) at 200 kV using a Cs-probe-corrected transmission electron microscope (FEI Titan Themis 200). Powders in solutions were diluted in ethanol and ultrasonicated. The colloid was placed on a TEM copper grid with carbon film and dried for several hours.

For rapid control of the size distributions of investigated NPs in aqueous solutions and their temporal stability, a dark-field laser ultramicroscope developed by us was used. Laser radiation (*λ* = 405 nm) was focused into the observed liquid volume in the form of a thin plate several microns thick (“light sheet”), located in the lateral direction relative to the microscopic objective. The solution of synthesized NPs in deionized water with NP concentration in the range of 10^−6^ ÷ 10^−4^ g/mL^3^ was placed in a transparent optical cuvette with transverse dimensions of 10 × 10 mm to a height of no more than 5–6 mm (to reduce the temperature induced convection of the liquid). Using a sensitive CCD-camera (PCO Sensicam em) operating in the continuous shooting mode, the microscopic images of single NPs and theirs aggregates entering the light sheet region due to random Brownian motion were continuously recorded as a sequence of video frames. The high sensitivity of the developed ultramicroscope made it possible to visualize single NPs as small as ~20 nm in size and to observe their individual trajectories of Brownian motion in aqueous solution. Figure 2 shows one of the video frames with images formed on the developed ultramicroscope of the synthesized 0.1 at. % Nd^3+^: LaF_3_ NPs in an aqueous solution, which fall into the “light sheet” area.

The special software determined the coordinates of the centers of gravity of the registered single NP images and, on this basis, defined their individual trajectories in a lateral plane for each time interval between successive frames. The subsequent analysis of the found trajectories allowed us to calculate the mean-squared displacement ${\bar{d}}^{2}$ for each particle in the lateral plane over time *t*, which related to their hydrodynamic radius *R_h_* via the Stokes–Einstein equation; this is valid for spherical particles and in the two-dimensional case (as it takes place in the case of observation under a microscope). The formula is as follows (see, e.g., [29])
(10)$${\bar{d}}^{2}=\frac{2K_{B}Tt}{3\pi R_{h}\eta }$$
where ${\bar{d}}^{2}$ is the mean-squared displacement of a particle with a hydrodynamic radius *R_h_* in the lateral plane over time t, *K_B_*—the Boltzmann constant, *T*—temperature and *η*—dynamic viscosity of liquid medium. Our measurements were carried out at *T* = 22.5 °C, the value of the viscosity of water was set equal to 0.943 mPa·s, and the time interval between successive video frames was 76.30 ms. Analysis of the detected NP images allowed us also to obtain information about the concentration of NPs. It is important to note that in the case of particles having a non-spherical shape and a complex structure, the value of their hydrodynamic radius determined using the NTA should be considered as a certain effective parameter characterizing the size and shape of such particles [30].

To determine the stability of the synthesized colloidal solutions of the 0.1 at. % Nd^3+^: LaF_3_ NPs, the zeta potential was measured using a Zetasizer Nano ZS (Malvern Instruments Ltd., Worcestershire, UK) at 25 °C based on laser Doppler velocimetry (λ = 633 nm). Prior to these measurements, a colloidal solution of investigated NPs with a concentration of ~20 mg/mL was preliminarily homogenized for 10 min using an ultrasonic bath.

### 3.3. Spectroscopic Research Methods

The study of the fluorescence kinetics of colloidal solutions of 0.1 at. % Nd^3+^: LaF_3_ NPs in NIR spectral range at different concentrations of Nd^3+^ ions was carried out using a pulsed tunable Ti–Sapphire laser (LS-2134-LT40, LOTIS TII, Minsk, Belarus) for fluorescence excitation (*t*_p_ = 8−30 ns, *f* = 10 Hz). An MDR-23 monochromator was used for the dispersion of fluorescence. To limit the incident laser radiation on the photodetector, an interference filter FEL0850 (Thorlabs, Newton, NJ, USA) was placed in front of the entrance slit of the monochromator. A Hamamatsu R13456P photomultiplier tube (PMT) operating in the single-photon counting mode was used as a radiation detector. The luminescence kinetics was recorded by the time-gated method of single photon counting using a TimeHarp 260 Nano multichannel scaler (PicoQuant, Berlin, Germany) with a specified nanosecond time gate duration. For precise synchronization of the detection system with the laser pulse, an avalanche pin photodiode was used. The electronic modules of the NIM standard were used as constant-fraction discriminators for start and stop pulses.

## 4. Results

### 4.1. Characterization of NPs with TEM

TEM studies of synthesized 0.1 at. % Nd^3+^: LaF_3_ NPs, deposited from an aqueous solution onto a TEM copper grid and dried for several hours, have shown that most single nanocrystals, which we will call primary NPs, are combined into clusters that can reach over 100 nanometers in size, having a complex fractal-like structure. Figure 3 demonstrates examples of the detected TEM images. Note that in these images we observed clusters located on the TEM grid, the size and fractal-like structure of which were able to change during drying.

The effect of aggregation of NPs in solutions into fractal-like structures, particularly the aggregation of colloidal NPs, has been observed and studied in a number of works (see, e.g., [23] and references therein). To characterize the complex and random spatial structure of fractal-like clusters, their fractal (Hausdorff) dimension ***D***_f_ is widely accepted, which quantifies how densely packed their constituent particles are. The number of primary particles *N* in a cluster (or the mass *m* of a cluster) increases with its effective size *R* (gyration radius *R_g_*) according to the non-integer power law [23,31]
(11)$$N=k_{f}{\left(\frac{R_{g}}{r}\right)}^{D_{f}},$$
where *k_f_* is the scaling prefactor, having a value between 1 and 1.2, and *r*—the radius of primary particle.

In general, particles can aggregate into porous clusters with different *D_f_*, which can range between 1 and 3. When *D_f_* is close to 1, clusters look like filamentous structures. If, on the other hand, *D_f_* is close to 3, tightly packed compact clusters are formed [23]. The parameters of aggregated NPs in solution are determined by the balance between the forces of attraction and repulsion, which are respectively determined by van der Waals and electrostatic interactions, sizes and concentration of primary NPs, and other parameters. In most cases, colloidal NPs driven by Brownian motion aggregate into clusters in two different modes. If they stick together at almost every collision, the aggregation processes is fast and called diffusion-limited cluster aggregation (DLCA). Under such conditions, clusters are formed, characterized by the value of *D_f_* in the range from 1.75 to 1.85 [32]. If, on the other hand, modification of the double layer surrounding the NPs drastically reduces their sticking probability, the slow or reaction limited cluster aggregation (RLCA) regime is entered, and more compact clusters with *D_f_* values between 2.0 and 2.1 are formed [32].

To describe the structure and packing density of primary particles in clusters, the ratio of gyration radius to hydrodynamic radius *R_g_*/*R_h_* is often used [33]. For clusters characterized by the presence of a loose structure and divergent branches in good solvent, this value is higher and can reach a value of over 2. For densely packed clusters close in shape to a sphere, this ratio approaches 0.8 [34]. In the case of clusters having a fractal structure, this ratio should be less than 1 ([30]).

To characterize the sizes of the observed primary NPs and structure of clusters based on them, we calculated two distributions: the distribution of the effective radii for each primary NP, regardless of whether it is included in the cluster or not, and the distribution of gyration radii for each cluster. To determine the value of effective radius of a primary NP, the image of each NP was replaced by a disk of equal area, the radius of which was used as effective radius. To calculate radii of gyration, we used the standard formula
(12)$$R_{g}=\sqrt{\frac{1}{2N^{2}}{\sum_{i=1}^{N}\sum_{j=1}^{N}\left({\bar{r}}_{i}-{\bar{r}}_{j}\right)}^{2}},$$
where *N*—number of primary NPs in cluster, and ${\bar{r}}_{i}$—coordinates of primary NPs. Note that the calculation of the gyration radius distribution, according to the above formula, is correct for a two-dimensional image, obtained using a TEM microscope. Therefore, the resulting distribution was converted to actual 3D-values using a constant conversion factor of 1.24 [35].

The resulting distributions are shown in Figure 4 together with the distribution of hydrodynamic radii, obtained on the basis of the NTA method.

### 4.2. Size Distribution Analysis of Synthesized Colloidal Nanoparticles

Let us analyze the obtained distributions of the sizes of primary NPs and the clusters formed by them. As we have already pointed out, both parameters under consideration, gyration and hydrodynamic radius, can be considered as parameters characterizing the effective sizes of clusters, although they are based on their different properties. The gyration radius reflects the spatial distribution of masses of primary particles in the cluster, while the latter—their joint hydrodynamic resistance. It should also be taken into account that in the case of NTA studies, we observe particles in aquatic environment; whereas in the case of TEM images, we detect clusters smeared after drying on a flat surface, the geometric parameters of which could change during drying. Therefore, the distributions under discussion should not completely coincide. Indeed, the distributions of the hydrodynamic and gyration radii shown in Figure 4, although similar in shape and characterized by close values of the parameters, do not coincide: the distribution curve of the hydrodynamic radii is clearly shifted towards larger values compared to the distribution curve of gyration radii.

The reason that the values of the hydrodynamic radii of the studied clusters exceed the values of the gyration radii requires an explanation. The ratio between the hydrodynamic and gyration radii of a cluster strongly depends on its structure. In the case of clusters consisting of approximately identical primary particles with a pronounced fractal structure, the values of the hydrodynamic radius, according to a number of theoretical studies (see, for example, [30]), should be less than the values of the gyration radius. Clusters consisting of approximately identical primary particles with a fractal structure can be roughly represented as loose coils limited in space, the density of which decreases with increasing distance from the center of the cluster.

Large values of hydrodynamic radii compared to the values of their gyration radii, as determined in our case, were observed for clusters with a structure other than fractal (see, for example [34]). The main difference between the clusters mentioned above was that they were characterized by the presence of “divergent branches”, which apparently caused an increase in the hydrodynamic resistance of these particles. As is known, a double electric layer is formed on the surface of colloidal NPs and in their immediate vicinity, which consists of a thin layer of immobile ions adjoining the surface and a wider region of mobile ions interacting with them. Therefore, it is not surprising that the hydrodynamic resistance of clusters consisting of colloidal NPs, the structure of which is characterized by the presence of “divergent branches”, may be greater than their gyration radius.

A more detailed understanding of this complex phenomenon is important for a more correct comparison of the results obtained in studies of colloidal NPs by electron microscopy and NTA. This makes relevant further research in this direction, which is accordingly planned by us in subsequent studies.

Thus, measuring the ratio of the hydrodynamic radius to the gyration radius of colloidal clusters provides important information about their structure. In the case when the hydrodynamic radius is less than the gyration radius, we can assume that the clusters have a fractal-like structure when the primary particles are grouped into some loose lumps limited in space. If the hydrodynamic radius exceeds the gyration radius, we can assume that the clusters are better described by a more complex structure, characterized by the presence of branches.

The main conclusion that can be drawn from the analysis of the data obtained by two fundamentally different methods, TEM and NTA, when studying colloidal nanoparticles in two different media, is that they provide information on the characteristic sizes and structure of the synthesized nanoparticles and clusters formed by them, as well as about the temporal stability of solutions of such particles. At the same time, the NTA method is simpler and much faster, which is very important for most applications.

### 4.3. Temporal Stability of NP Colloidal Solutions

An important issue for applications of colloidal NPs is the stability of their solution parameters over time. One of the processes leading to undesirable changes in these parameters is the aggregation of NPs, which can cause coagulation of a colloidal solution. Stability of colloidal solutions, as is known, is largely determined by the value of the zeta potential. Our measurement of the zeta potential gave a value of +47 mV ± 5% that is rather high and means that the synthesized colloidal solution should be stable.

To control the temporal stability of the 0.1 at. % Nd^3+^: LaF_3_NPs colloidal solution and obtain additional information about the aggregation processes, we measured the size distribution in a freshly prepared solution (three days after preparation) and in a solution subjected to long-term storage (seven months) in a stationary state. The measurements were performed in two modes: (a) without preliminary preparation of the solution and (b) after ultrasonic treatment of the solution. Results of these measurements are shown in Figure 5.

It can be seen that long-term storage of a solution of colloidal 0.1 at. % Nd^3+^: LaF_3_NPs leads to a shift of the distribution maximum towards larger sizes and an increase in the distribution width. This shows that during a long storage, some of the NPs stacked together. Sample homogenization in an ultrasonic bath made it possible to significantly eliminate this undesirable effect.

The measurements also showed that, after long-term storage of synthesized colloidal solutions in a vertical tube, the size distribution of NPs in the upper and lower parts of the tube began to differ (Figure 6). Under the sedimentation, the proportion of larger nanoclusters increased in the lower part of the cell and decreased in the upper part.

The slow process of NP agglomeration in the studied colloidal solutions during storage can be explained by the adhesion of clusters, when they come into contact during convection and Brownian motion. It is significant that in our case, this process could be partially reversed by using ultrasonic treatment of the solution. From this fact and the very slow coagulation of the solution during storage, it can be assumed that the main part of the primary NPs had already rapidly aggregated during the synthesis and isolation of the desired fraction. Note that a detailed analysis of the TEM data obtained with the high spatial resolution showed that the crystallization process in each primary nanocrystal occured independently of each other. Thus, further studies are required for a deeper understanding of the aggregation process of the studied NPs.

### 4.4. Fluorescence Quenching Kinetics and Radiative Lifetime of the ^4^F_3/2_ Level of the Nd^3+^ Ion

The fluorescence decay kinetics of the metastable ^4^F_3/2_ level of the Nd^3+^ ion in the 0.1 at.% Nd^3+^: LaF_3_ NPs in aqueous and DMSO media was measured at the ^4^F_3/2_→^4^I_9/2_ transition of Nd^3+^ at λ*_det_* = 863 nm upon excitation to the ^4^F_5/2_ + ^2^H_9/2_ mixed state at λ*_exc_* = 789 nm (Figure 7). The fluorescence kinetics of the 0.1 at. % Nd^3+^: LaF_3_ NPs dissolved in DMSO (Figure 7, red curve), and ACS-DMSOx2 NPs (Figure 7, magenta curve) decays more slowly at the initial stage than the kinetics of NPs in any of the aqueous solutions: ACS colloid (Figure 7, blue curve) and DMSO-ACS colloid (Figure 7, green curve). The DMSO-ACS colloid exhibits reversible luminescent properties with respect to the initial ACS colloid, since the fluorescence kinetics of these samples (Figure 7, blue and green curves) is very similar. At the same time, the fluorescence kinetics of the ACS-DMSOx2 colloid (Figure 7, magenta curve) obtained by double redispersion of NPs in DMSO decays more slowly compared to the kinetics of the DMSO colloid redispersed only once (Figure 7, red curve).

Since the concentration of Nd^3+^ ions is very low, one can neglect both the contribution of Nd–Nd self-quenching to the decay of the measured luminescence kinetics, and the migration of electronic excitation over neodymium ions to both types of acceptors to unexcited neodymium ions and OH- molecular groups. Thus, the measured fluorescence kinetics can be represented as a product of two terms: the kinetics of static quenching $N_{0}\left(t\right)$ of excited Nd^3+^ donors on OH- acceptors (Equation (4)) and the kinetics of spontaneous emission
(13)$$I_{meas}\left(t\right)=N_{0}\left(t\right)\exp\left(-\frac{t}{\tau_{R}}\right).$$

To determine the relative fluorescence quantum yield φ (Equations (6) and (7)), it is necessary to know the value of the spontaneous radiative lifetime $\tau_{R}$ of the ^4^F_3/2_ excited state of the Nd^3+^ ion. The method to find the radiative lifetime for ACS colloid in the case of static fluorescence quenching was described in detail in [28]. The value of $\tau_{R}$ = 1300 μs was found there. In the case of DMSO solvent, the kinetics of impurity quenching is also the Förster kinetics of Equation (4), and the kinetics of fluorescence decay is described by Equations (4), (5) and (13) with $\gamma_{A}$*=*
$\gamma_{OH}$. Next, we need to analyze Equation (13) in more detail. To determine $\tau_{R}$, it is necessary to construct a function depending on $\sqrt{t}$. For each chosen value of $\tau_{R}$, we then analyze the correspondence of the experimental data to a straight line. With the optimal selection of the value of $\tau_{R}$, we should obtain a straight line over the entire measurement interval, from the slope of which we can determine the value of the static quenching macroparameter $\gamma_{OH}$
(14)$$\ln\left[N_{0}\left(t\right)\right]=\ln\left[I_{meas}\left(t\right)/\exp\left(-\frac{t}{\tau_{R}}\right)\right]$$

After correctly selecting the $\tau_{R}$ = 930 µs for DMSO and ACS-DMSOx2 colloids (Figure 8, red and magenta curves, respectively), the impurity fluorescence quenching kinetics gives an exactly straight line with a slope equal to $\gamma_{OH}$. The refractive index of the organic medium DMSO $n_{med}\left(\mathrm{DMSO}\right)$ = 1.477 [36] is higher than the refractive index of water $n_{med}\left(\mathrm{water}\right)$ = 1.333. Therefore, according to [37], the radiative lifetime of the NPs dissolved in DMSO is shorter, and the rate of spontaneous emission is higher than those dissolved in water.

According to Equation (5) the ratio of macroparameters $\gamma_{OH}\left(\mathrm{DMSO}\right)/\gamma_{OH}\left(\mathrm{ACS}\right)$ (Figure 8, blue and red curves, respectively) is equal to the ratio of concentrations of OH- groups in the NPs, i.e., $n_{OH}\left(\mathrm{DMSO}\right)/n_{OH}\left(\mathrm{ACS}\right)$ = 0.025/0.043 = 0.58. Thus, we found that the concentration of OH- groups inside the NPs dispersed in DMSO is almost 40% lower than in NPs dispersed in aqueous medium. The concentration of the OH- groups in the ACS-DMSOx2 NPs is even less. It is determined by the ratio of $\gamma_{OH}\left(\mathrm{ACS}-\mathrm{DMSOx}2\right)/\gamma_{OH}\left(\mathrm{ACS}\right)$ (Figure 8, blue and magenta curves, respectively), which gives $n_{OH}\left(\mathrm{ACS}-\mathrm{DMSOx}2\right)/n_{OH}\left(\mathrm{ACS}\right)$ = 0.023/0.043 = 0.53.

### 4.5. Relative Fluorescence Quantum Yield

The dependences of relative fluorescence quantum yield φ on concentration of OH- groups were calculated for all studied colloidal solutions using Equations (6) and (7) (Figure 9). For this, the value of the microparameter of donor–acceptor interaction $C_{DA}\left(\mathrm{Nd}-\mathrm{OH}\right)$ = 0.0056 nm^6^/ms [28] was taken. To determine the value of $C_{DA}\left(\mathrm{Nd}-\mathrm{OH}\right)$, it is necessary to know not only the slope of the Förster stage (Equation (5)), but also the rate of the initial ordered stage and the boundary time $t_{bound}$ between them [28]. The experimentally-found radiative lifetimes of the ^4^F_3/2_ level, $\tau_{R}$ = 1300 μs for the ACS colloid, and $\tau_{R}$ = 930 μs for a solutions of NPs dissolved in DMSO solutions, were taken for calculations. The difference between the two theoretical curves for ACS and DMSO colloids is due to different radiative lifetimes. As follows from the figure, this difference increases with an increase in the concentration of acceptors. However, a more significant increase in the quantum yield is obtained after a decrease in the concentration of acceptors in the DMSO colloid.

The experimental values of the relative fluorescence quantum yield φ for measured colloidal solutions using Equation (1) and their calculated values using Equation (6) were found. It can be concluded that the measured and calculated values coincide with an accuracy of 2% (Table 1). This means that the static quenching model used to calculate φ is valid for the studied NPs. This makes it possible to determine the concentrations ratio of OH- groups (Table 1) in the volume of NPs for various colloidal solutions by superimposing the experimentally-found values of the relative fluorescence quantum yields φ on the theoretical concentration dependences. By using Equations (6)–(9) and the value of microparameter $C_{DA}\left(\mathrm{Nd}-\mathrm{OH}\right)$ = 0.0056 nm^6^/ms [28], we also determined the absolute concentration of OH- groups ($n_{OH}$) in the volume of nanoparticles in the colloids (Table 1).

The concentration of OH- acceptors $n_{OH}$ can be determined much more accurately from the slope of the kinetics of impurity quenching using Equation (5), the experimentally-measured macroparameter $\gamma_{OH}$ (Figure 8) and the microparameter of donor–acceptor interaction $C_{DA}\left(\mathrm{Nd}-\mathrm{OH}\right)$. The discrepancy within 5% in the values of $n_{OH}$ concentrations (Table 1) obtained by two methods is mainly due to the error in the experimentally-found values of the relative fluorescence quantum yield φ using Equation (1). This error arises due to an inaccurately determined area under the luminescence decay kinetics curve, the initial stage of which (from 0 to several microseconds) is distorted due to the contribution of the scattered laser excitation light resulting from the low concentration of Nd^3+^ ions and the contribution of upconversion. Due to the distortion of the initial stage of the luminescence decay kinetics, it is also impossible to properly normalize the decay curve to the luminescence maximum. At the same time, the determination of the macroparameter $\gamma_{OH}$ is much more reliable, because it is determined by the slope of the Förster stage of fluorescence impurity quenching kinetics $N_{0}\left(t\right)$ (Equation (4)) throughout its entire length. Moreover, the slope of the Förster kinetics in the coordinates $\ln\left(N_{0}\left(t\right)\right)$ vs. $t^{1/2}$ does not depend on the normalization of the luminescence decay kinetics.

It has been established that concentration of OH- acceptors ($n_{OH}$ = 1.28–2.33 nm^−3^, Table 1) is approximately two orders of magnitude higher than the concentration of the Nd^3+^ acceptors ($n_{Nd}$ = 1.76 10^−2^ nm^−3^). This result is consistent with the small contribution of Nd–Nd self-quenching and energy migration over the Nd^3+^ ions to the fluorescence quenching.

Thus, the fluorescence quantum yield φ almost doubled when the solvent was changed from water to DMSO (Table 1). This is explained by the fact that when the NPs dispersion medium is changed from water to DMSO, some of the acceptors (about a half) are removed from the volume of NPs. The second reason for the increase in the quantum yield is due to the decrease in the radiative lifetime $\tau_{R}$. As follows from Formula (7), the argument x decreases by half with a corresponding decrease in the concentration of acceptors, and decreases by a factor of $\sqrt{930/1300}\approx 0.85$ due to a decrease in $\tau_{R}$. The final decrease in the argument x by a factor of 0.42 led to an almost twofold increase in the quantum yield. Therefore, the Nd^3+^: LaF_3_ NPs dispersed in DMSO solvent can be promising for biomedical application.

### 4.6. *Thermogravimetric Analysis* and Differential Scanning Calorimetry Results

To test the effect of DMSO on Nd^3+^: LaF_3_ NPs precipitated from an aqueous solution, they were analyzed by differential scanning calorimetry (DSC) with weight analysis of the 0.1 at. % Nd^3+^: LaF_3_ powders obtained by drying at 80 °C an aqueous colloid and DMSO colloid of nanoparticles (Figure 10). Analysis of gravimetric curves (TG) demonstrates that for NPs obtained in aqueous solution (Figure 10, blue curves) without additional processing steps, there is a slight decrease in weight (up to 0.5 wt.%), associated with the removal of physically and chemically bound water from the surface and from the pores of the NPs.

This is confirmed by numerous endothermic effects at about 45 °C, 190 °C, 385 °C, 435 °C and a smooth endothermic course of the DSC curve starting from 500 °C. In the case of NPs treated in DMSO (Figure 10, red curves), a similar course of the TG curve up to 100 °C is observed, due to the removal of physically adsorbed moisture from the surface of NPs, which is confirmed by similar data on DSC (Figure 10, bottom graph).

With increasing temperature, a more significant change in weight is observed, associated with the removal of water and DMSO from the particles under study. In the temperature range of 350–400 °C, a sharp decrease in weight is observed, associated with the complete removal of DMSO from the surface and from the pores of NPs due to their destruction. Further weight change is negligible.

The DSC curve of the DMSO-treated sample (Figure 10, lower figure, red curve) is similar to the DSC curve of the sample (ACS) without post-treatment (Figure 10, lower figure, blue curve) except for the range of 500–600 °C. This temperature interval during heating corresponds to the removal of chemically bound anions from the surface of NPs, and in the case of NPs obtained from aqueous solutions, it corresponds to dehydration, i.e., removal of hydroxyl ions. The difference in the DSC curves (Figure 10, blue and red curves) shows that this dehydration process is not observed in the sample without post-treatment in DMSO. The presence of such a process during post-treatment with DMSO indicates that DMSO actively interacts with hydroxyl ions on the surface and in the pores of NPs, removing them due to the good solubility of water in DMSO, which reduces the concentration of OH- groups that quench the luminescence of Nd^3+^ ions in LaF_3_ NPs.

The decrease in the weight of the powder of particles obtained from an aqueous colloid of NPs without post-treatment upon heating to 800 °C (Figure 10, upper graph, blue curve) is about 0.5 wt.%. At the same time, the mass content of hydroxyl ions in the same nanoparticles in the initial aqueous colloid, determined from the analysis of the kinetics of impurity luminescence quenching, is about 1.15 wt.% (Table 1). A possible explanation why not all water “leaves” the particles when they are heated to 800 °C is that the hydroxyl ions inside the NP volume at temperatures above 600 °C undergo pyrohydrolysis with the incorporation of oxygen into the LaF_3_ crystal lattice. The possibility of the presence of oxygen in single-phase LaF_3_ samples is confirmed by the data of [38,39], in which samples of RF_3_ xH_2_O (R=La, Nd) were melted in a vacuum of 10^−2^ mm Hg and, according to the results of chemical analysis, 0.010–0.015 and 0.4–0.5 wt.% of the oxygen content were found, respectively.

At the same time, the weight loss of the powder particles obtained from the colloid with post-treatment in DMSO at 800 °C is about 1.4 wt.% (Figure 10, top figure, red curve), which is two times larger than the mass concentration of OH- groups (*n*_OH_ (in DMSO) = 0.69 mass%), determined from the analysis of the kinetics of impurity luminescence quenching in the DMSO colloid of these NPs (Table 1). The reason for this discrepancy in the concentration of water in NPs, which was determined by two methods, is that, when the powder is heated, not only part of the OH- groups, but also DMSO molecules are volatilized from the particles.

## 5. Conclusions

In this research, three samples of 0.1 at. % Nd^3+^: LaF_3_ NPs colloidal solutions prepared by HTMW treatment were studied in different media, one in water and two in DMSO. The fluorescence static quenching kinetics of NPs $N_{0}\left(t\right)$ was found for each of the colloidal solutions from the measured fluorescence kinetics and analyzed in the coordinates $\ln\left(N_{0}\left(t\right)\right)$ versus $t^{1/2}$. In doing so, the radiative lifetime $\tau_{R}$ for each colloidal solution was determined simultaneously with the determination of the macroparameter of static quenching $\gamma_{OH}$ (Equation (5)). We found that concentration of OH- quenching acceptors in the volume of the NPs in the DMSO colloidal solutions is almost twice lower than in the aqueous one. At the same time, the radiative lifetime $\tau_{R}$ decreases in DMSO solutions due to its higher refractive index compared to water. Together, both these effects lead to a decrease in the dimensionless argument x (Equation (7)) by a factor of 0.42. This gives an almost two-fold increase in the fluorescence quantum yield φ of the NPs in DMSO colloidal solutions compared to the aqueous one. It can be concluded that the experimental values of φ found using Equation (1) coincide with the theoretical calculated ones using Equations (6) and (7) with an accuracy of 2%. This made it possible to determine the concentration of the OH- groups in the volume of NPs in two types of colloids using the theoretical calculated dependences of φ on their concentration.

At the same time, the concentration of acceptors can be determined much more accurately from the analysis of the kinetics of impurity fluorescence quenching. As shown in [28], for this it is necessary to know not only the slope of the Förster stage (Equation (5)), but also the rate of the initial ordered $W_{ord}$ stage and the boundary time $t_{bound}$ between them: this allows the value of microparameter $C_{DA}\left(\mathrm{Nd}-\mathrm{OH}\right)$ = 0.0056 nm^6^/ms [28] and thus the absolute value of acceptor concentration to be found. The discrepancy in the values of *n*_OH_ concentrations obtained by these two methods lies within 5% (Table 1). It should be noted that the concentrations of acceptors in the studied systems lie in the range of $n_{OH}$ = 1.28–2.33 nm^−3^, which is two orders of magnitude higher that the concentration of Nd^3+^ ions. Thus, the static mechanism of the fluorescence quenching was confirmed.

The dimensions and structure of the synthesized colloidal clusters of 0.1 at. % Nd^3+^: LaF_3_ NPs were studied according to TEM data and optical scattering ultramicroscopy, operating at an unshifted frequency. The combined application of two fundamentally different methods made it possible to conclude that the increased hydrodynamic viscosity of the formed clusters is better described by the presence of a loose structure with divergent side branches than by a fractal structure.

Long-term studies of colloidal solutions by the NTA method have shown that NPs slowly aggregate during long-term storage, and under the action of ultrasound this process can be reversed to a significant extent, which indicates the weakness of the bonds formed. The irreversible fraction of observed clusters arises either during the synthesis or shortly after it. According to TEM data with high spatial resolution, the joint growth of nanocrystals during the synthesis is not observed.

## Figures and Tables

**Figure 1 nanomaterials-12-03749-f001:**
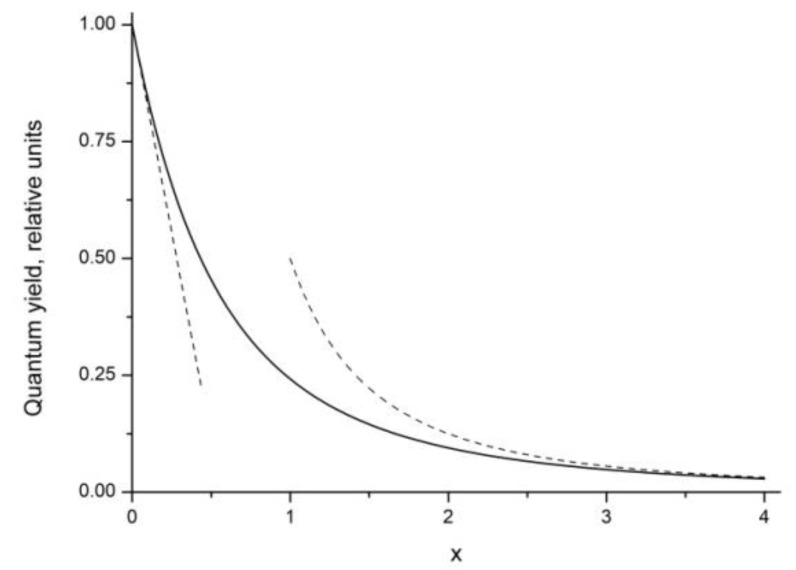
Fluorescence relative quantum yield φ of the static quenching Equation (6) as a function of the argument $x=\gamma_{A}\sqrt{\tau_{R}}/2$. Dashed lines are calculated using approximate Formulas (8) and (9).

**Figure 2 nanomaterials-12-03749-f002:**
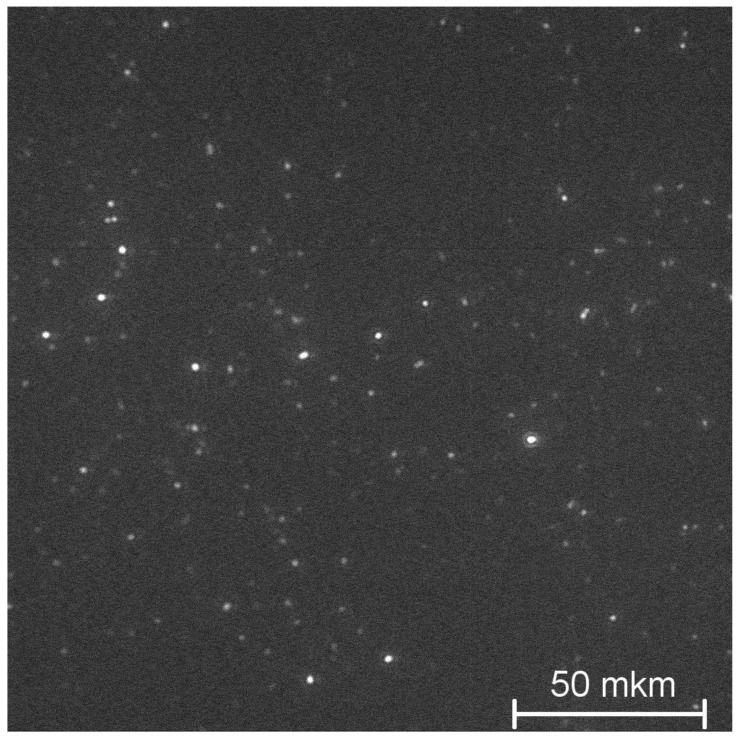
Example of individual images of the 0.1 at. % Nd^3+^: LaF_3_ nanoparticles in an aqueous solution observed with the developed ultramicroscope.

**Figure 3 nanomaterials-12-03749-f003:**
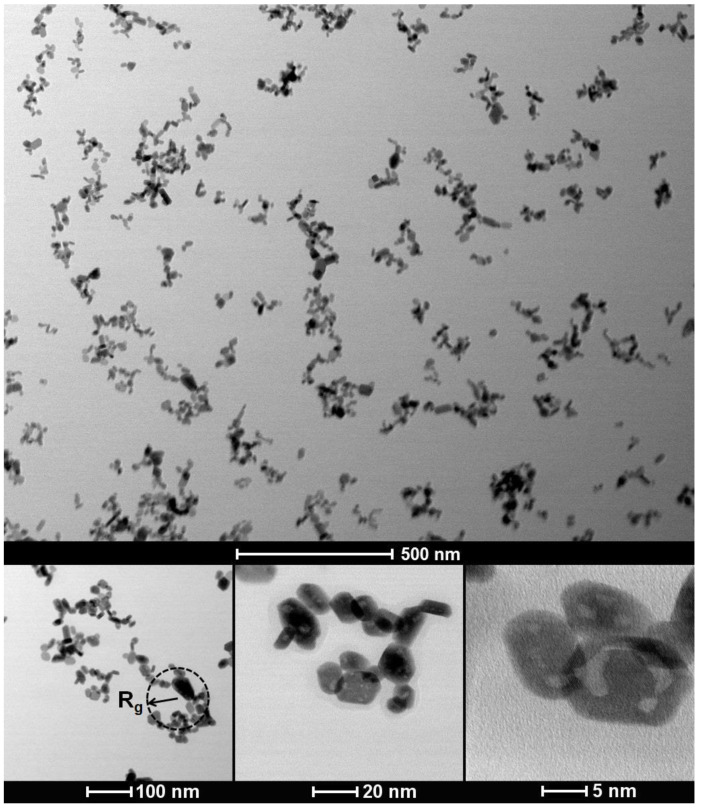
TEM images of 0.1 at. % Nd^3+^: LaF_3_ nanoparticles and clusters formed by them placed on a TEM copper grid with carbon film and dried for several hours. At the bottom left of the figure, an example of the calculated gyration radius for 2D image is drawn in black.

**Figure 4 nanomaterials-12-03749-f004:**
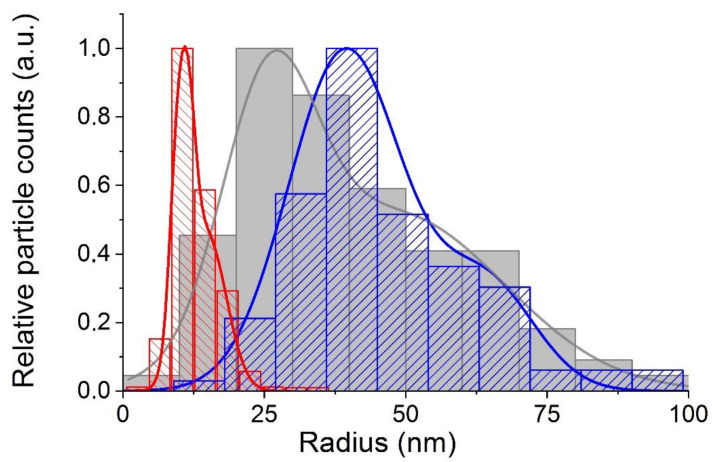
Histograms and distributions characterizing the sizes of synthesized colloidal nanocrystals and clusters formed by them. The narrowest distribution (red) and distribution showing intermediate sizes (grey) were obtained from the analysis of TEM images of particles deposited on a TEM copper grid with carbon film, whereas the distribution showing the largest sizes (blue) were obtained on the basis of analysis of individual trajectories of Brownian motion of nanoparticles and their clusters dispersed in purified deionized water. The distribution with the narrowest width was calculated for sizes of primary NPs, without taking into account their aggregation into clusters, while the distribution showing the largest sizes demonstrates the values of individual hydrodynamic radius of clusters in aqua solution. The distribution with the intermediate sizes was calculated for clusters. It was converted to actual 3D-values and shows distribution of their individual gyration radii.

**Figure 5 nanomaterials-12-03749-f005:**
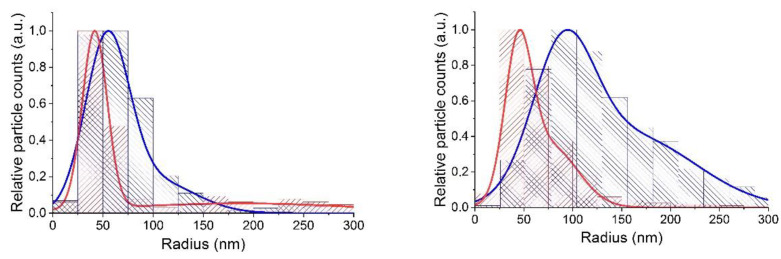
Size distributions of the colloidal solution of LaF_3_:Nd^3+^ nanoparticles in deionized water, measured with the ultramicroscope, 3 days (**left panel**) and 7 months (**right panel**) after sample preparation. Blue—data obtained without applying ultrasonic treatment to the colloidal solution, red—after 20 min of such exposure.

**Figure 6 nanomaterials-12-03749-f006:**
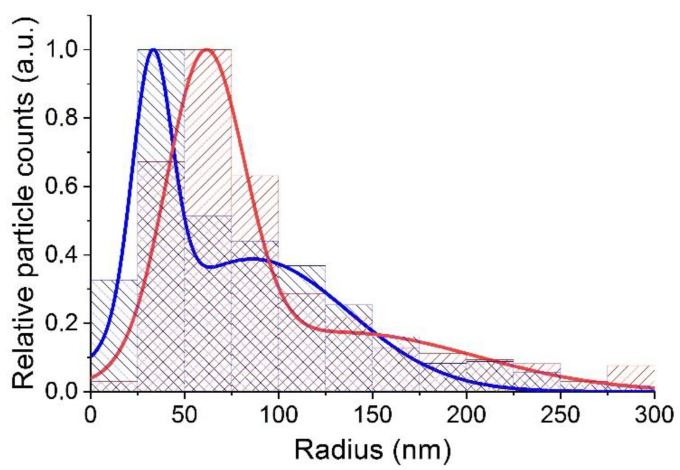
The size distributions of the 0.1 at. % Nd^3+^: LaF_3_ nanoparticles in deionized water as measured for samples taken from the upper (blue) and lower (red) parts of the cell.

**Figure 7 nanomaterials-12-03749-f007:**
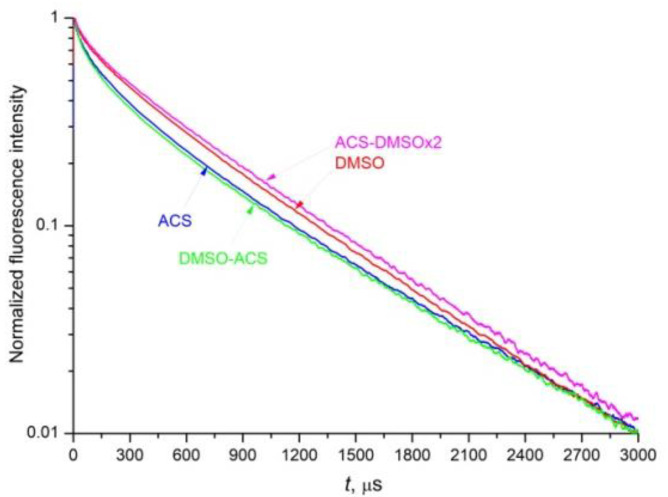
Kinetics of fluorescence decay of the ^4^F_3/2_ level of the Nd^3+^ ion of the 0.1 at.% Nd^3+^: LaF_3_ NPs under laser excitation at a wavelength of 789 nm and detection of fluorescence at a wavelength of 863 nm—in aqueous colloidal solutions: ACS and DMSO-ACS colloids (blue and green curves, respectively); and in organic DMSO solvent colloidal solutions: DMSO and ACS-DMSOx2 colloids (red and magenta curves, respectively).

**Figure 8 nanomaterials-12-03749-f008:**
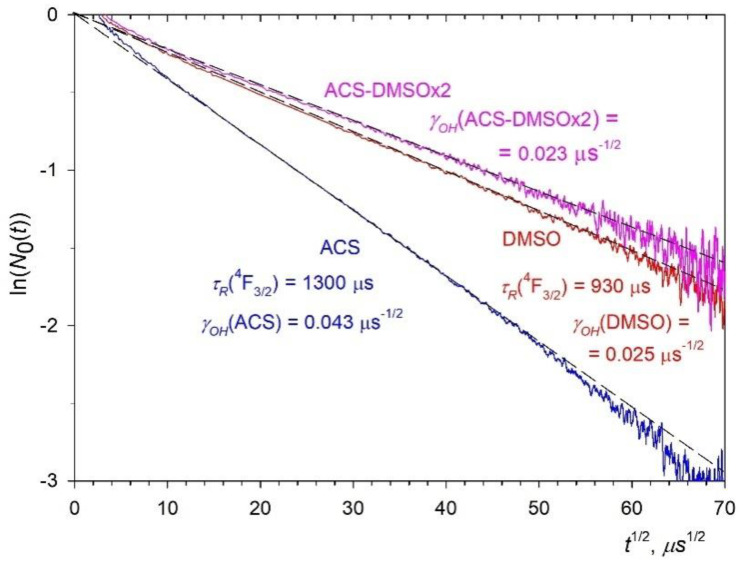
Kinetics of static fluorescence quenching of the ^4^F_3/2_ level of the Nd^3+^ ion of the 0.1 at. % Nd^3+^: LaF_3_ NPs colloids in special coordinates: ACS—blue curve, DMSO—red curve, and ACS-DMSOx2—magenta curve; dashed thin black lines show the slope of the impurity fluorescence quenching kinetics, which is used to determine the energy transfer macroparameter $\gamma_{OH}$.

**Figure 9 nanomaterials-12-03749-f009:**
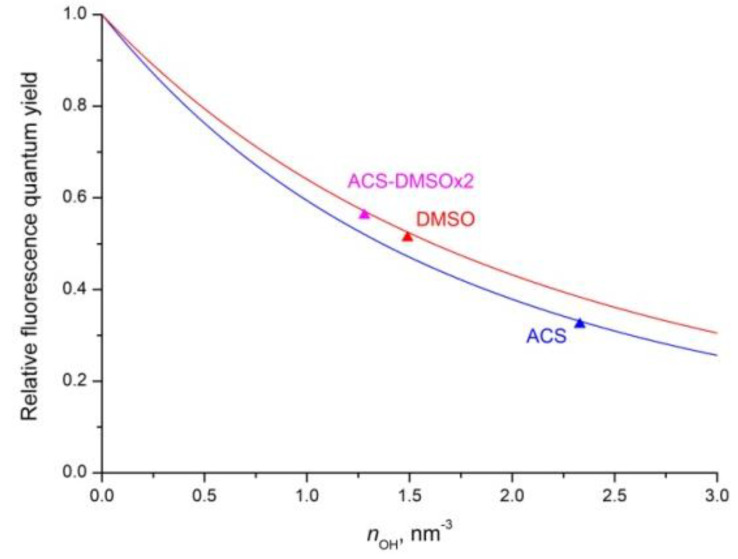
Calculated dependences of the relative fluorescence quantum yield φ of colloidal solutions of the 0.1 at. % Nd^3+^: LaF_3_ NPs on the concentration of OH- groups in the bulk of NPs: ACS colloid—blue curve, and DMSO colloid—red curve; triangles are the experimentally measured values of the fluorescence quantum yield (see Table 1).

**Figure 10 nanomaterials-12-03749-f010:**
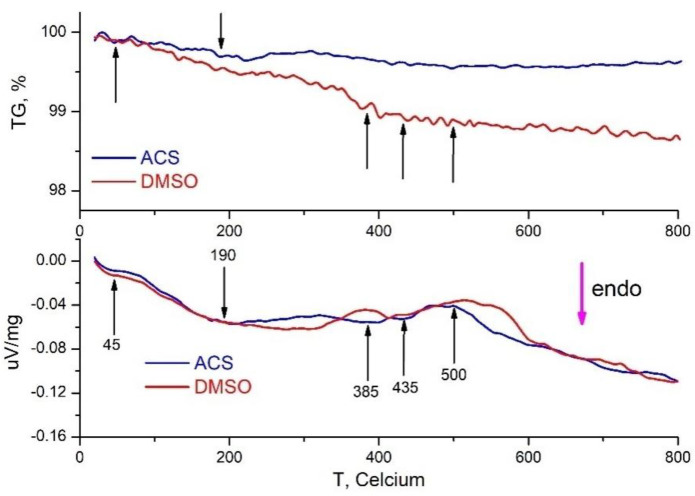
Differential scanning calorimetry (DSC) (lower figure) with synchronized thermogravimetric analysis (upper figure) of the 0.1 at. % Nd^3+^:LaF_3_ nanoparticle powders obtained by drying at 80 °C aqueous (ACS, blue curves) and DMSO (DMSO, red curves) colloids of nanoparticles.

**Table 1 nanomaterials-12-03749-t001:** Relative fluorescence quantum yield φ and concentration of OH- groups inside the NPs for three colloidal solutions of 0.1 at. % Nd^3+^: LaF_3_ NPs.

Solvent	*φ*, rel. Units Calculated	*γ_OH_*, µs^−1/2^ Measured	*φ*, rel. Units Measured	*n_OH_*, nm^−3^ from *φ* Meas	*n_OH_*, nm^−3^ from *γ_OH_*
Aqua	0.331	0.043	0.324	2.33	2.45
DMSO	0.515	0.025	0.513	1.49	1.42
DMSOx2	0.571	0.023	0.562	1.28	1.31

## Data Availability

Not applicable.

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
