# Peer review of "The Influence of Medium on Fluorescence Quenching of Colloidal Solutions of the Nd3+: LaF3 Nanoparticles Prepared with HTMW Treatment"

_nanomaterials, 2022, doi:10.3390/nano12213749_

Round 1
Reviewer 1 Report
it is a nice experimental study which should be followed up by theoretical studies (on the level of DFT-studies/Coupled Cluster)
Author Response
Thank you very much for appreciating of our manuscript. We will think about DFT computations of coupled clusters in our future studies. In addition, we plan to study the concentration dependence of the fluorescence intensity and kinetics in these coupled clusters of NPs.
Reviewer 2 Report
In the manuscript “The influence of solvent on fluorescence quenching of colloidal solutions of the Nd3+: LaF3 nanoparticles prepared with HTMW treatment“ the authors report the synthesis of three samples of NPs colloidal solutions, 0.1 at. % Nd3+: LaF3 by HTMW treatment in different solvents, in the water and two in the DMSO.
The fluorescence static quenching kinetics of NPs was found for each of the colloidal solutions from the measured fluorescence kinetics and analyzed.
The sizes of synthesized NPs were monitored by common TEM and by applying a rapid procedure based on optical visualization of the trajectories of the Brownian motion of NPs in solution using a laser ultramicroscope.
The use combined application of two different methods allowed obtaining important information regarding dimensions and structure of the synthesized colloidal clusters of 0.1 at. % Nd3+: LaF3 NPs.Overall, the analyses of the data are adequate for the above-mentioned issue.
The conclusions are relevant and supported by the results, being of interest for the readership of Nanomaterials Journal.
Accordingly, I consider tthis appropriate for publication, taking into account the next comments and suggestions.
1. The bibliography of the manuscript need to be updated. The most recent works cited are from 2017 2. The authors claim that from analysis of TEM image fig 3 of the synthesized colloidal clusters of 0.1 at. % Nd3+:LaF3 NPs, “ have shown a complex fractal-like stru The fractal behaviour can be investigated by TEM micrographs analysis. The D0 fractal dimensions and the lacunarities of the nanoparticles can be computed using the “box counting” method and the modified black and white TEM images. fig. 3 (Reference: Characterization of bimetallic nanoparticles by fractal analysis. Powder Technology, 338, 905–914. doi:10.1016/j.powtec.2018.07.083). How to varies the fractal dimension of the synthesized colloidal clusters of 0.1 at. % Nd3+: LaF3 NPs in in different solvents. 3. Please check the 27 reference.
Author Response
- The bibliography of the manuscript need to be updated. The most recent works cited are from 2017.
- The authors claim that from analysis of TEM image Fig 3 of the synthesized colloidal clusters of 0.1 at. % Nd3+:LaF3 NPs, “have shown a complex fractal-like structure. The fractal behaviour can be investigated by TEM micrographs analysis. The D0 fractal dimensions and the lacunarities of the nanoparticles can be computed using the “box counting” method and the modified black and white TEM images. Fig. 3 (Reference: Characterization of bimetallic nanoparticles by fractal analysis. Powder Technology, 338, 905–914. doi:10.1016/j.powtec.2018.07.083). How to varies the fractal dimension of the synthesized colloidal clusters of 0.1 at. % Nd3+: LaF3 NPs in different solvents.
Yes, indeed, we wrote in the article that the studied clusters observed with TEM looks similar to fractals. However, the results of our measurements of the distributions of the gyration and hydrodynamic radii of the investigated clusters did not confirm their fractal nature. Therefore, we believe that the analysis of the fractal dimension of clusters of nanoparticles synthesized by us does not make much sense. A brief description of fractal clusters given in the article was presented only to improve understanding of the material.
More details:
The ratio of gyration radius to hydrodynamic radius Rg/Rh is often used to describe the structure and packing density of primary particles in clusters. For clusters characterized by the presence of a loose structure and divergent branches in good solvent this value is higher and can reach a value of 2 and more. For densely packed clusters close in shape to a sphere, this ratio approaches 0.8 (Burchard, 1999). In the case of clusters having a fractal structure, this ratio should be less than one ([30] Lattuada M., Wu H., Morbidelli M.).
The results of our measurements of the distributions of the gyration and hydrodynamic radii for the clusters of NPs formed in solution showed that the distribution of gyration radii turned out to be shifted towards smaller values, compared with the distribution of hydrodynamic radii. This result contradicts the predictions of the theory developed to describe the Brownian motion of fractal particles. Thus, our experimental results do not confirm the fractal nature of the clusters we study.
Considering the comments made and in order to make the content of the article more understandable, we have introduced a number of small additions to the text and made several minor changes
- The article (reference [29]) has been replaced with a new one (2020).
- New text and new reference [33] were inserted to page 8 (highlighted in yellow):
To describe the structure and packing density of primary particles in clusters the ratio of gyration radius to hydrodynamic radius Rg/Rh is often used [33]. For clusters characterized by the presence of a loose structure and divergent branches in good solvent this value is higher and can reach a value of 2 and more. For densely packed clusters close in shape to a sphere, this ratio approaches 0.8 [34]. In the case of clusters having a fractal structure, this ratio should be less than one ([30]).
and:
Note that the calculation of the gyration radius distribution, according to the above formula, is correct for a two-dimensional image, obtained using a TEM microscope. Therefore, the resulting distribution was converted to actual 3D-values using a constant conversion factor of 1.24 [35].
- The Figure 3 has been replaced with a new one, where instead of the distribution of the “two-dimensional” values of the gyration radius, the distribution for “the three-dimensional” values of the gyration radii is shown.
- The last sentence in the caption to Figure 3 has been replaced with two new sentences:
The distribution with the intermediate sizes was calculated for clusters. It was converted to actual 3D-values and shows distribution of their individual gyration radii.
- Please check the 27 reference.
We have changed the Ref. 27 to
Handbook of Mathematical Functions with Formulas, Graphs and Mathematical Tables. Ed. M. Abramowitz and I. Stegun. National Bureau of Standards, Applied Mathematics Series 55, Issued June 1964.
Reviewer 3 Report
I read the paper titled "The influence of solvent on fluorescence quenching of colloidal solutions of the Nd3+: LaF3 nanoparticles prepared with HTMW treatment"
I found the research interesting because it provides a new opportunity for scientists involved in luminescent materials to reduce the quenching of the NIR fluorescence of colloidal solutions of 0.1 at. % Nd3+: LaF3 nanoparticles .
The paper is well-written and the goal is focused and well-realized by an accurate description of the technique and data interpretation.
The manuscript is potentially interesting for readers of Nanomaterials. As a reviewer, I still have some comments and suggestions.
1. I suggest to be careful with the word dissolved (lines 157 and 161) considering that the nanoparticles are not soluble in the solvents, dispersed should be better and more reliable
check typos such as line 343 (subscript in the formula)
2. Please improve the quality of the figure,
- Figure 1 correct a.u. (atomic unit) with arb. unit
- Figure 10 corrects the x ax label unit Celsius
Author Response
- I suggest to be careful with the word dissolved (lines 157 and 161) considering that the nanoparticles are not soluble in the solvents, dispersed should be better and more reliable
Thank you for very fruitful comments.
We have changed the word dissolved to dispersed as follows.
Next, this colloidal aqueous solution of NPs was divided into two equal parts, one part of the NPs was precipitated by centrifugation and dissolved dispersed in the same volume of DMSO (the condition was strictly observed that the volume concentration of NPs in both colloid samples was the same – 20 mg/ml). As a result, two samples were obtained: an aqueous colloidal solution of 0.1 at. % Nd3+: LaF3 NPs (ACS) and 0.1 at. % Nd3+: LaF3 NPs in a DMSO dispersion medium (DMSO). The DMSO colloid was then centrifuged again and redissolved re-dispersed in water to obtain a reduced aqueous colloidal solution of LaF3 NPs (DMSO-ACS).
check typos such as line 343 (subscript in the formula)
Unfortunately, our numeration of the lines is different. Therefore, we have checked subscripts in all formulas.
2. We have corrected Fig. 1 and Fig. 10.
Reviewer 4 Report
This paper reports the fluorescence property of Nd3+: LaF3 nanoparticles prepared with HTMW treatment using different solvents.First, I am concerned that no information on the crystal structure of these NPs is given. In order for the discussion in the text to be valid, it is necessary to confirm that the powder diffraction patterns of these two samples are almost the same.Fluorescence properties are considered to be strongly influenced by the crystallinity of the sample.
The authors need to clearly establish whether the difference in fluorescence properties is really due only to the concentrations of OH-.
Therefore, I judged that a major revision is required for this paper.
Author Response
First, I am concerned that no information on the crystal structure of these NPs is given. In order for the discussion in the text to be valid, it is necessary to confirm that the powder diffraction patterns of these two samples are almost the same. Fluorescence properties are considered to be strongly influenced by the crystallinity of the sample.
The authors need to clearly establish whether the difference in fluorescence properties is really due only to the concentrations of OH-.
Thanks to the reviewer for this valuable comment. Since NPs were not synthesized in DMSO but only redispersed in it, the crystal structure did not change. X-ray phase analysis of these NPs is presented in [28]. We have changed the text in the manuscript accordingly and the title:
A colloidal solution of 0.1 at. % Nd3+: LaF3 NPs was synthesized from an aqueous solution by the hydrothermal method with microwave treatment (HTMW). Its detailed characterization including X-ray phase analysis of the synthesized LaF3 NPs was done carried out in Ref. [28] (see Supplementary Materials).
New title of the manuscript: The influence of solvent medium on fluorescence quenching of colloidal solutions of the Nd3+: LaF3 nanoparticles prepared with HTMW treatment
As concerned with the last comment we think that we explained rather clearly the difference in fluorescence properties. See the text from the manuscript below.
Thus, the fluorescence quantum yield increased almost twice when the solvent was changed from water to DMSO (Table 1). This is explained by the fact that when the NPs dispersion medium is changed from water to DMSO, part of the acceptors (about a half) is removed from the volume of NPs. The second reason for the increase in the quantum yield is due to the decrease in the radiative lifetime . As follows from formula (7), the argument decreases by half with a corresponding decrease in the concentration of acceptors and decreases by a factor of due to a decrease in . The final decrease in the argument by a factor of 0.42 led to an almost twofold increase in the quantum yield.
Round 2
Reviewer 4 Report
I am satisfied wit the authors' revision.
Now it is ready for publication.